# Current out of pocket care costs among HIV and hypertension co-morbid patients in urban and peri-urban Uganda

**Drew B. Cameron** [1‡*], **Lillian C. Morrell** [2‡], **Faith Kagoya** [3], **John Baptist Kiggundu**[3], **Brian Hutchinson**[2], **Robert Twine**[3], **Jeremy I. Schwartz**[4,5], **Martin Muddu**[6], **Gerald Mutungi**[7], **James Kayima**[8,9], **Anne R. Katahoire**[10], **Chris T. Longenecker**[11], **Rachel Nugent**[2], **David Contreras Loya**[12], **Fred C. Semitala**[3,6,9]

**1** Department of Health Policy and Management, Yale School of Public Health, New Haven, Connecticut, United States of America, **2** Center for Global Noncommunicable Diseases, RTI International, Seattle, Washington, United States of America, **3** Infectious Disease Research Collaboration, Kampala, Uganda, **4** Section of General Internal Medicine, Yale School of Medicine, New Haven, Connecticut, United States of America, **5** Uganda Initiative for Integrated Management of Non-Communicable Diseases, Kampala, Uganda, **6** Makerere University Joint AIDS Program, Kampala, Uganda, **7** Uganda Ministry of Health, Kampala, Uganda, **8** Uganda Heart Institute, Kampala, Uganda, **9** Department of Internal Medicine, Makerere University Kampala, Kampala, Uganda, **10** Child Health and Development Centre, Makerere University College of Health Sciences, School of Medicine, Kampala, Uganda, **11** Division of Cardiology, Department of Global Health, University of Washington, Seattle, Washington, United States of America, **12** Tecnológico de Monterrey, Monterrey, México

‡ DBC and LCM contributed equally to this work as co-first authors.
* drew.cameron@yale.edu

**Data Availability Statement:** All underlying data files for this manuscript are available from the

## Abstract

### Background

Despite improvements to the cascade of HIV care in East Africa, access to care for non-communicable disease co-morbidities like hypertension (HTN) remains a persistent problem. The integration of care for these conditions presents an opportunity to achieve efficiencies in delivery as well as decrease overall costs for patients. This study aims to build evidence on the burden of current out-of-pocket costs of care among HIV-HTN co-morbid patients.

### Methods

We administered a pre-tested, cross-sectional, out-of-pocket cost survey to 94 co-morbid patients receiving HIV care from 10 clinics in the Wakiso and Kampala districts of Uganda from June to November 2021. The survey assessed socio-demographic characteristics, direct medical costs (e.g., medications, consultations), indirect costs (e.g., transport, food, caregiving), and economic costs (i.e., foregone income) associated with seeking HIV and HTN care, as well as possible predictors of monthly care costs. Patients were sampled both during a government-imposed nation-wide full COVID-19 lockdown (n = 30) and after it was partially lifted (n = 64).

Harvard Dataverse: https://doi.org/10.7910/DVN/NMEQ0M.

**Funding:** Research reported in this publication was supported by the National Heart, Lung, And Blood Institute of the National Institutes of Health (https://www.nhlbi.nih.gov) under Award Number UG3HL154501 (awarded to authors FS and CTL). The content is solely the responsibility of the authors and does not necessarily represent the official views of the National Institutes of Health. The study funders had no role in study design, collection, analysis, interpretation of data, writing of the report or decision to submit for publication.

**Competing interests:** The authors have declared that no competing interest exist.

## Results

Median HIV care costs constitute between 2.7 and 4.0% of median monthly household income, while HTN care costs are between 7.1 to 7.9%. For just under half of our sample, the median monthly cost of HTN care is more than 10% of household income, and more than a quarter of patients report borrowing money or selling assets to cover costs. We observe uniformly lower reported costs of care for both conditions under full COVID-19 lockdown, suggesting that access to care was limited. The main predictors of monthly HIV and HTN care costs varied by disease and costing perspective.

## Conclusions

Patient out of pocket costs of care for HIV and HTN were substantial, but significantly lower during the 2021 full COVID-19 lockdown in Uganda. New strategies such as service integration need to be explored to reduce these costs.

## Introduction

In Uganda, strong infrastructure and funding have driven major improvements in the HIV care cascade (i.e. rates of diagnosis, treatment, and viral suppression) [1]. As of 2021, 89% of people living with HIV (PLHIV) in Uganda knew their status, 92% of those received antiretroviral therapy, and 95% of those had achieved viral suppression [2]. However, as the population ages, a growing body of evidence finds significant non-communicable disease (NCD) co-morbidity among PLHIV. Despite a national policy of no-cost provision of direct care costs, including NCD medications supplied at no cost to patients by the National Medical Store, drug stockouts are common and access to care for these conditions among PLHIV in Uganda remains suboptimal [3–10].

During the formative stages of PULESA-Uganda, an ongoing stepped-wedge cluster randomized trial in Uganda (NCT05609513), we evaluated the out-of-pocket (OOP) costs of HIV and hypertension (HTN) care to co-morbid patients. HTN significantly increases cardiovascular disease risks among PLHIV, yet comprehensive literature on the OOP costs associated with managing HIV and HTN in low- and middle-income countries remains limited [11].

To fill this evidence gap, we pilot-tested an out-of-pocket (OOP) costs survey questionnaire in the districts of Kampala and Wakiso. This OOP cost survey was part of a formative study which evaluated knowledge, attitudes, and perceptions towards HTN management as well as perceived barriers and facilitators to integration of HIV and HTN care in ten health facilities. The OOP survey aimed to collect data on patient demographics, to estimate and describe personal out-of-pocket costs for HIV and HTN care for PLHIV. The survey also contained questions prompting respondents to report *direct costs* paid for services (e.g., consultation fees, medications), *indirect costs* to access services (e.g., transportation, child/elder care, and food or other expenses) and *opportunity costs* (i.e., lost wages due to work time missed to access care) separately for HIV and HTN.

One of the aims of the PULESA-Uganda trial is to evaluate two combinations of strategies to integrate care for HIV and HTN. This survey collected data on access to and the costs of HIV and HTN care prior to formal integration of HIV and HTN care through the PULESA-Uganda trial, and so provides insight into potential inefficiencies in care access and cost that may have been present prior to integration of these services. The current perception is that

most comorbid patients manage their HIV and HTN at different clinics and separate clinical visits, though in practice little is known about current care seeking behavior. Current literature suggests that HTN treatment initiation, 1-year retention, and control are suboptimal among co-morbid patients and demonstrate notable quality gaps [4, 5]. Co-locating and/or synchronizing service delivery for multiple conditions can theoretically reduce the number of required clinic visits, creating efficiencies that manifest as savings in health-related patient expenditures. Results of this survey would help to inform the future testing of such a hypothesis. These results would also inform the subsequent economic model design and input parameters for the PULESA-Uganda trial.

We provide the results of this pilot out-of-pocket survey to gain insight into the financial and economic burden on PLHIV and HTN and to understand the potential efficiencies and inefficiencies that might result from care integration. We add to the scarce literature on patient costs to access care for multiple conditions in Uganda by reporting results from the survey as disaggregated direct, indirect, and opportunity costs attributable to the management of each disease. Our results will inform planning and implementation of integrated programs, and the ongoing design of policies to reduce major costs that act as barriers to care.

## Methods

The main reporting elements, as outlined in the Consolidated Health Economic Evaluation Reporting Standards (CHEERS) checklist, can be found in the S1 Checklist [12].

### Study design

We conducted a cross-sectional survey to estimate out-of-pocket (OOP) costs associated with accessing care among PLHIV and HTN in ten HIV clinics in urban Kampala and peri-urban Wakiso districts of central Uganda. The OOP survey was adapted from a survey developed previously for use in Cote d'Ivoire and consisted of a pre-tested questionnaire to estimate and describe the out-of-pocket cost expenditures of PLHIV and HTN [13]. The original survey collected only financial out of pocket costs and was adapted significantly for the local context in Uganda and to capture a more expansive set of economic out of pocket costs faced by patients. The survey contained questions on each person's most recent medical visits for each condition (for HIV, the most recent visit was that day's visit; for HTN, patients were asked to recall their last visit for care, which took place before the date of the survey within the last 12 months). Data were collected on social, demographic, and household characteristics, direct medical costs (such as HTN medications and registration fees), indirect non-medical costs (e.g., transportation), and opportunity costs (i.e., foregone income).

### Study setting

We purposefully selected a sample of clinics to conduct formative research in advance of the PULESA-Uganda trial. In planning our project, we obtained a comprehensive list of all HIV clinics in both Wakiso and Kampala districts. After review, we determined that clinics serving fewer than 400 patients per year would be heterogenous and difficult to study. Among the remaining 60+ clinics serving more than 400 patients, the median clinic size was roughly 2,000 patients per year. We sought to include an equal number of small (<2,000 patients) vs. large (>2,000 patients) clinics so that study findings could be more generalizable. Further, we believed that the larger HIV clinics would be more likely to have resources not available to smaller ones. We also aimed to include a mix of public and private/not-for-profit facilities. After stratifying clinics by relative size and type across the two districts, we selected ten facilities (five each in Wakiso and Kampala districts) in which to administer the formative

**Table 1. Study sites and number of participants enrolled in the KAP and OOP surveys.**

| Name and type of facility* | No. PLWHIV | Facility type | KAP survey | OOP survey |
|---|---|---|---|---|
| **Kampala district** | | | | |
| Kisenyi HC IV | 11,972 | Public | 118 | 27 |
| Kawaala HC III | 8,814 | Public | 86 | 21 |
| Komamboga HC III | 4,732 | Public | 30 | 8 |
| Nsangi HCIV | 1,865 | Public | 18 | 4 |
| Butabika Hospital | 1,310 | Public | 13 | 3 |
| **Wakiso district** | | | | |
| St. Francis Hospital, Nsambya | 7,911 | PNFP | 78 | 19 |
| Kisubi Hospital | 2,221 | PNFP | 22 | 5 |
| Kira HC IV | 1,120 | Public | 14 | 3 |
| Nakawuka HC III | 852 | Public | 7 | 2 |
| Kawanda HC III | 830 | Public | 8 | 2 |
| **Total** | *41,627* | | **394** | **94** |

Notes: Abbreviations: **HC** = Health Center; **KAP** = Knowledge, Attitudes, and Practice survey; **OOP** = Out-of-pocket; **PNFP** = Private-not-for-profit. *The Ugandan Health Care system is a 7-tier health system; at the bottom is the community level which includes Community Health workers who provide basic health education and preventive services. Health Center II provides a wide range of health care services including immunization, maternal and childcare services, outpatient services, and basic laboratory services. Health Centre III is located at the county level and offers a broader range of primary care services including general outpatient care, inpatient care, deliveries, basic laboratory care, and surgeries. Health Centre IV serves as a referral center at the district level and provides more comprehensive in-patient and out-patient services including surgery, emergency obstetric care, laboratory services, and radiology. At the 5th level are general hospitals which are district-level facilities that provide serve as referral centers for lower-level facilities and offer advanced diagnostic services and specialized medical and surgical services, emergency services, and intensive care units. Level 7 are the National Referral Hospitals, providing highly specialized medical care, advanced surgery, and specialized diagnostic services and serve as centers for medical training and research. Level 6 are the regional referral hospitals.

assessments (see **Table 1**). A different combination of 16 clinics was later chosen for inclusion in the PULESA-Uganda trial.

## Participant selection and recruitment

From 23 June 2021 through 25 November 2021, we enrolled 394 participants in a Knowledge, Attitudes, and Perceptions (KAP) survey—the main component of the formative assessment. We piggy-backed the OOP expenditure survey on the larger KAP survey to reduce client burden and survey costs. Written informed consent was obtained from all participants before their inclusion in the survey. The participants were identified through review of existing paper-based and electronic medical records as well as referrals from health care providers. In each of the ten HIV clinics, a list of all PLHIV was generated, and a convenience sample was consecutively enrolled until target sample sizes were reached for each facility. At the time of study planning, we did not know how many patients per facility were co-morbid (with HIV and HTN). Sample size per facility was determined based on estimated HTN prevalence within the population [14]. Participants were required to 1) be aged ≥18 years, 2) have been diagnosed with HIV and enrolled in treatment, 3) have had a hypertension diagnosis (defined either by BP >140/90 on two or more occasions during the preceding 12 months, or use of anti-hypertensive medication), and 4) be willing to provide informed consent. We excluded those who met the inclusion criteria but had a) advanced cognitive impairment or another condition that severely impaired their ability to complete study assessments and/or, b) who used anti-hypertensive medication solely for an indication other than hypertension (e.g., systolic heart failure / heart failure with reduced ejection fraction). For the OOP survey, we selected every fourth participant who completed the KAP survey (starting with KAP

participant #4) at each of the study sites to limit survey fatigue, arriving at a random subsample of 94 participants. All facilities were proportionally represented in the study based on the total number of HIV patients that was known at the time of study planning, with roughly 0.2 percent of each population of PLHIV (who were also co-morbid with HTN) from each clinic included in the study. As this study was exploratory / hypothesis generating, there were no ex-ante power calculations conducted.

## Data collection

The OOP survey was administered to 94 participants, following administration of the KAP survey on the same day. Trained research assistants conducted translated surveys in Luganda, the most spoken language in the Central region of Uganda. Data were collected using the Research Electronic Data Capture system (REDCap). All cost data were collected in 2021 Ugandan Shillings (UGS). Following Kumaranayake (2000), price data were then inflated to 2022 UGS using the World Bank's GDP Price Deflator, and then exchanged to 2022 USD using an exchange rate of 3,587.05UGS / $1USD [15–17].

## Data analysis

Data were analyzed using Stata v.18. We first generate basic descriptive statistics to character-ize our sample, comparing the sample taken during a 2021 full COVID-19 lockdown to that taken after the lockdown was partially lifted. Then, we enumerate the direct, indirect, and opportunity costs of care incurred by PLHIV and HTN on a per-visit basis. Next, we examine the self-reported total number of monthly visits for each condition, as well as monthly spend-ing for each as a percent of household income and expenditure. Finally, we examine potential predictors of both "financial" and "economic" out-of-pocket costs of monthly care (multiply-ing the last care visit by the number of projected visits per month) for each disease among our sample of co-morbid patients. For both HIV and HTN monthly care costs, "financial" cost measures included direct and indirect out of pocket costs extrapolated from the last visit taken for care. "Economic" costs include the cost of foregone labor from the last care visit in addition to financial costs. For monthly hypertension costs, we calculate the cost of pharmaceutical drugs based on patient monthly self-report data rather than drugs purchased during last visit (see: **S1 Appendix** for more details).

In our analysis of predictors of monthly cost, we model financial and economic costs for both conditions as a function of a set of key demographic characteristics as well as visit- and condition-specific explanatory variables. We use a generalized linear models (GLM) estima-tion, a maximum likelihood generalization of the ordinary least squares approach allowing for added flexibility in the assumption of the error variance distribution to account for heteroske-dasticity. We assume a log-link function and Gaussian probability distribution. We estimate the following primary *model 1* specification for stated patient HIV out of pocket financial costs (FC):

$$ln(HIV_{FC})_{ic} = \beta_0 + \beta_1 lockdown_i + \beta_2 \log(visits)_{ic} + \beta_3 admission_{ic} + \beta_4 care_{ic} + \beta_5 travel\ time_{ic} + \beta_6 transit\ mode_{ic} + \beta X_{ic} + \gamma_c + \epsilon_{ic} \tag{1}$$

where $ln(HIV_{FC})_{ij}$ is the financial monthly out of pocket HIV care cost for patient $i$ sampled at facility $c$, *lockdown* represents whether the survey took place during the COVID-19 full travel lockdown (1) or after it was partially lifted (0), $log(visits)$ represents the logged total number of monthly visits per respondent as a direct input to overall costs, *admission* represents whether the respondent has been admitted overnight to a hospital facility for disease-specific complica-tions in the last 12 months, *care* represents any care provision made for adults or children as a

result of the visit, *travel time* represents the combined number of hours required to travel to and from the care visit, *transit mode* is a categorical variable for the mode of transport taken to the HIV facility (including *matatu* (mini bus), private car, private motorcycle, *boda-boda* (motorcycle taxi), or multi-modal transport, with walking/biking as the omitted reference category), $X$ is a vector of respondent-specific demographic covariates including age, gender, education, employment, monthly household expenditures and marital status, and $\gamma$ is a vector of facility fixed effects.

$$ln(HIV_{EC})_{ic} = \beta_0 + \beta_1 lockdown_i + \beta_2 \log(visits)_{ic} + \beta_3 admission_{ic} + \beta_4 care_{ic} + \beta_5 travel\ time_{ic} \\ + \beta_6 transit\ mode_{ic} + \beta_7 foregone\ income_{ic} + \beta X_{ic} + \gamma_c + \epsilon_{ic} \tag{2}$$

*Model 2* examines economic costs $ln(HIV_{EC})_{ic}$ for patient $i$ at facility $c$, including any stated lost income from attending HIV visits in the dependent monthly HIV out of pocket cost estimates. Right-hand-side variables in model 2 are the same as model 1 but additionally include a binary covariate *foregone income* to indicate whether the patient reported foregoing any income-generating activities to attend the last HIV visit. For each HIV outcome in models 1 and 2, we show nested results both without and with the $X$ vector of respondent-specific demographic covariates to examine model fit.

$$ln(HTN_{FC})_{ic} = \beta_0 + \beta_1 lockdown_i + \beta_2 \log(visits)_{ic} + \beta_3 admission_{ic} + \beta_4 care_{ic} + \beta_5 travel\ time_{ic} \\ + \beta_6 same\ facility_{ic} + \beta X_{ic} + \epsilon_{ic} \tag{3}$$

For monthly hypertension costs in *model 3* we repeat the same empirical strategy as that taken for HIV in model 1. The dependent variable $ln(HTN_{FC})_{ic}$ is financial HTN care costs for patient $i$ sampled at HIV facility $c$ but additionally includes monthly self-reported hospitalization costs normalized over a 1-year period. Because data on HTN care is based on a prior visit for which we have limited information (including no mode of transportation), we do not include transportation-related covariates only available for HIV data. However, we do include an additional binary covariate *same facility* to indicate whether the patient self-reports attending the same HIV clinic for their HTN care versus a different facility/location.

$$ln(HTN_{EC})_{ic} = \beta_0 + \beta_1 lockdown_i + \beta_2 \log(visits)_{ic} + \beta_3 admission_{ic} + \beta_4 care_{ic} + \beta_5 travel\ time_{ic} \\ + \beta_6 same\ facility_{ic} + \beta_7 foregone\ income_{ic} + \beta X_{ic} + \gamma_c + \epsilon \tag{4}$$

*In model 4*, we examine the economic costs of care $ln(HTN_{EC})_{ic}$ including foregone income from labor for patient $i$ sampled at facility $c$ and, in a departure from Model 3, add a covariate for whether the patient gave up any income to attend HTN visits (*foregone income*).

For models 1–4 we carefully considered the inclusion of additional clinic-level covariates such as facility size and facility type (public vs. private/not-for-profit). In our HIV models (1 and 2) direct HIV costs that would be realized at a given facility constituted a vanishingly small share of total costs of care, insufficient in our estimation to justify the inclusion of additional descriptive covariates for these facilities. Further, we do not have reason to assume that characteristics like patient load or facility type would have any relationship to the predominately indirect and opportunity costs of HIV care included in the dependent variable beyond what should be captured by including a vector ($\gamma_c$) of facility fixed effects. Thus, these variables were not included to avoid concerns about model over-fit. In models 3 and 4 we did not have any data on the HTN facilities themselves visited for HTN care to include additional facility-level covariates such as clinic size or type.

### Ethics approval and consent to participate

Ethical approval for the study was obtained from the Makerere University School of Medicine Research and Ethics Committee (REC REF Mak-SOMREC-2021-58), the Uganda National Council for Science and Technology (SS808ES), and the Yale University Institutional Review Board (IRB Protocol #: 2000030432). The study also obtained administrative clearance to conduct research from the Uganda Ministry of Health, Kampala Capital City Authority, and Wakiso District Local Government. Written informed consent was obtained from all participants before their inclusion in the survey. Confidentiality and anonymity of participants' data were maintained throughout the study.

### Inclusivity in global research

Additional information regarding the ethical, cultural, and scientific considerations specific to inclusivity in global research is included in the S1 Checklist.

## Results

Of the 94 patients participating in the survey, the first 30 were surveyed under a nationwide, government-instituted 42-day full lockdown (June 18[th] to July 29[th], 2021) during the COVID-19 pandemic, which involved the closure of all educational institutions, some bans on travel (with the exception of patients traveling to medical facilities), night-time travel curfews after 7PM, as well as restrictions on open markets, church services, and many forms of social gathering. The lockdown was eased on 30 July 2021 with only partial restrictions on public transportation (half capacity) that remained in place until 30 December 2021 [18–21]. The remaining 62 participants were sampled during this easing of the lockdown. Baseline characteristics for individuals sampled using the OOP survey are shown in **Table 2,** both for the full sample (col 1) as well as comparing the set of patients surveyed under full lockdown (col 2) and under partial lockdown (col 3).

Females constitute 74% of those surveyed. Participants had a mean age of 52 and traveled an average of 16.6 kilometers (km) to get to the HIV facility for an appointment. Under full lockdown, patients traveled an average of 11.7 km versus 19.0 km under partial lockdown, though the mean difference in travel distance is not statistically significant. Patients reported taking half the time to travel to the HTN facility on average compared to the HIV facility. The average monthly household (HH) income of all surveyed patients was $153.55 USD. Patients reported an average monthly HH expenditure of $158.20 USD. Though the difference in both expenditures and income are not statistically significantly different between those sampled under full lockdown versus those measured after the lockdown was partially lifted, point estimates show a larger difference in average reported earnings (USD $53.40 lower under full lockdown) compared to expenditure (USD $15.06 lower under full lockdown) between the two groups. Around one-third (36.2%) of the sample population reported needing to pay for childcare for their HIV clinic visit and a little more than one-fifth (22.3%) reported needing to pay for childcare for their last HTN visit. Most patients (59.0%) paid for food while traveling to or from their HIV clinic visit. Over one-third (34.0%) of patients paid for food while traveling to or from their HTN clinic visit. Of all sampled patients, more than one-half (59.6%) lost wages or earnings because of their HIV clinic visit while only 37.2% lost wages or earnings because of their HTN visit.

**Table 3** shows the cost breakdowns per HIV and HTN visit for the full sample (col 1) and disaggregated by lockdown status (col 2 and col 3). A median total financial cost of $3.51 USD was observed for the HIV clinic visits in the full sample. A financial cost of $2.48 was found for the patients that attended the clinic under government full lockdown. Notably, none of the

**Table 2. Descriptive statistics (means) for full sample and by lockdown status.**

| | Full Sample (n = 94) | | | Full lockdown (n = 30) | | Partial lockdown (n = 64) | | |
|---|---|---|---|---|---|---|---|---|
| | Mean [SD] | n | | Mean (SE) | n | Mean (SE) | n | p-value |
| **DEMOGRAPHICS** | | | | | | | | |
| Age | 52.0 | 94 | | 52.5 | 30 | 51.8 | 64 | 0.750 |
| | [10.1] | | | (1.5) | | (1.4) | | |
| Female | 0.74 | 94 | | 0.73 | 30 | 0.75 | 64 | 0.865 |
| | [0.44] | | | (0.08) | | (0.06) | | |
| Married | 0.33 | 94 | | 0.27 | 30 | 0.36 | 64 | 0.378 |
| | [0.49] | | | (0.08) | | (0.06) | | |
| Secondary education or better | 0.36 | 94 | | 0.27 | 30 | 0.41 | 64 | 0.193 |
| | [0.48] | | | (0.08) | | (0.06) | | |
| Employed | 0.75 | 94 | | 0.77 | 30 | 0.75 | 64 | 0.863 |
| | [0.43] | | | (0.08) | | (0.06) | | |
| Distance to HIV facility (km) | 16.6 | 90 | | 11.7 | 30 | 19.0 | 60 | 0.204 |
| | [25.4] | | | (2.1) | | (3.9) | | |
| Time to HIV facility (hours) | 1.4 | 94 | | 1.2 | 30 | 1.5 | 64 | 0.461 |
| | [1.3] | | | (0.2) | | (0.2) | | |
| Time to HTN facility (hours) | 0.7 | 94 | | 0.5 | 30 | 0.8 | 64 | 0.112 |
| | [0.7] | | | (0.1) | | (0.1) | | |
| Patients who visit different facility for HTN care | 0.75 | 94 | | 0.90 | 30 | 0.67 | 64 | 0.018** |
| | [0.44] | | | (0.06) | | (0.06) | | |
| Monthly HH income (USD) | $153.55 | 83 | | $115.60 | 24 | $169.00 | 59 | 0.114 |
| | [$139.17] | | | ($15.25) | | ($20.31) | | |
| Monthly HH expenditure (USD) | $158.20 | 82 | | $148.10 | 27 | $163.16 | 55 | 0.714 |
| | [$173.12] | | | ($31.07) | | ($24.25) | | |
| **STATISTICS AT LAST CLINIC VISIT** | | | | | | | | |
| Patients who used childcare (HIV visit) | 0.36 | 94 | | 0.33 | 30 | 0.38 | 64 | 0.699 |
| | [0.48] | | | (0.09) | | (0.06) | | |
| Patients who used childcare (HTN visit) | 0.22 | 94 | | 0.17 | 30 | 0.25 | 64 | 0.371 |
| | [0.42] | | | (0.07) | | (0.05) | | |
| Patients who paid for food (HIV visit) | 0.59 | 94 | | 0.47 | 30 | 0.64 | 64 | 0.113 |
| | [0.50] | | | (0.09) | | (0.06) | | |
| Patients who paid for food (HTN visit) | 0.34 | 94 | | 0.27 | 30 | 0.38 | 64 | 0.307 |
| | [0.48] | | | (0.08) | | (0.06) | | |
| Patients who lost wages or earnings (HIV visit) | 0.60 | 94 | | 0.53 | 30 | 0.63 | 64 | 0.404 |
| | [0.49] | | | (0.09) | | (0.06) | | |
| Patients who lost wages or earnings (HTN visit) | 0.37 | 94 | | 0.30 | 30 | 0.41 | 64 | 0.326 |
| | [0.49] | | | (0.09) | | (0.06) | | |
| **WITHIN THE LAST 12 MONTHS** | | | | | | | | |
| Patients who had any hospitalization for HIV | 0.04 | 94 | | 0.10 | 30 | 0.02 | 64 | 0.060* |
| | [0.20] | | | (0.06) | | (0.02) | | |
| Patients who had any hospitalization for HTN | 0.10 | 94 | | 0.03 | 30 | 0.13 | 64 | 0.163 |
| | [0.30] | | | (0.03) | | (0.04) | | |

**Notes:** Standard deviations reported in brackets; Standard errors reported in parenthesis; Student's t-tests for each row examine mean difference between sample taken during full lockdown vs. partial lockdown; *** p<0.01, ** p<0.05, * p<0.1.

**Table 3. Median cost (USD) per visit breakdown.**

| | Full sample | Full lockdown | Partial lockdown |
|---|---|---|---|
| | Median [IQR] | Median [IQR] | Median [IQR] |
| HIV CARE | | | |
| *Total direct cost* | $0.00 | $0.00 | $0.00 |
| | [$0.00 - $0.00] | [$0.00 - $0.00] | [$0.00 - $0.00] |
| *Total indirect cost* | $3.36 | $2.48 | $3.80 |
| | [$2.05 - $6.43] | [$0.88 - $3.51] | [$2.41 - $7.48] |
| Transport costs (one-way) | $1.46 | $1.46 | $1.61 |
| | [$0.88 - $2.92] | [$0.00 –$2.34] | [$0.88 –$3.07] |
| HH care costs | $0.00 | $0.00 | $0.00 |
| | [$0.00 - $0.00] | [$0.00 –$0.00] | [0.00–0.00] |
| Additional food expenses | $0.58 | $0.00 | $0.73 |
| | [$0.00 - $0.88] | [$0.00 - $0.58] | [$0.00 - $0.88] |
| Other costs | $0.00 | $0.00 | $0.00 |
| | [$0.00 - $0.00] | [$0.00 - $0.00] | [$0.00 - $0.00] |
| *Total opportunity cost* | $1.90 | $1.61 | $2.92 |
| | [$0.00 - $5.85] | [$0.00 - $4.38] | [$0.00 - $6.58] |
| **Total (Direct + Indirect)** | $3.51 | $2.48 | $4.24 |
| **Total (Direct + Indirect + Opportunity)** | $6.43 | $4.24 | $9.21 |
| HTN CARE | | | |
| *Total direct cost* | $2.92 | $0.00 | $4.38 |
| | [$0.00 - $8.91] | [$0.00 - $4.97] | [$1.46 - $14.61] |
| *Total indirect cost* | $1.61 | $0.00 | $2.27 |
| | [$0.00 - $4.68] | [$0.00 - $2.92] | [$0.15 - $5.92] |
| Transport costs (one-way) | $0.58 | $0.00 | $0.73 |
| | [$0.00 - $1.75] | [$0.00 - $1.17] | [$0.00 - $2.78] |
| HH care costs | $0.00 | $0.00 | $0.00 |
| | [$0.00 - $0.00] | [$0.00 –$0.00] | [$0.00 - $0.00] |
| Additional food expenses | $0.00 | $0.00 | $0.00 |
| | [$0.00 - $0.58] | [$0.00–0.44] | [0.00–0.73] |
| Other costs | $0.00 | $0.00 | $0.00 |
| | [$0.00 - $0.00] | [$0.00 - $0.00] | [$0.00 - $0.00] |
| *Total opportunity cost* | $0.00 | $0.00 | $0.00 |
| | [$0.00 - $2.34] | [$0.00 - $0.73] | [$0.00 - $3.65] |
| **Total (Direct + Indirect)** | $5.70 | $2.92 | $6.94 |
| **Total (Direct + Indirect + Opportunity)** | $6.65 | $3.65 | $9.43 |
| **n** | **94** | **30** | **64** |

**Notes:** [$] Though not individually reported, total direct costs of care for HIV and HTN included costs related to facility user fees, consultation fees, laboratory and test fees, and drug costs; Interquartile ranges (IQR: 25th– 75th percentiles) are reported in brackets; Direct costs include facility user fees, consultation fees, laboratory and test fees, and drug costs; Other costs include cellular service fees, fines from law enforcement, and informal payments; Indirect cost totals and all financial and economic totals account for round-trip transportation costs. No missing values were observed for any of the reported cost inputs.

input cost categories are statistically significantly different for per-visit HIV care between participants sampled under full lockdown, and those sampled after. **Fig 1** shows the average cost per visit breakdown by cost category. Most of the total financial costs were from indirect costs such as transport costs, household care costs, additional food expenses, and other costs which included cellular service fees, fines from law enforcement, and informal payments. Transport

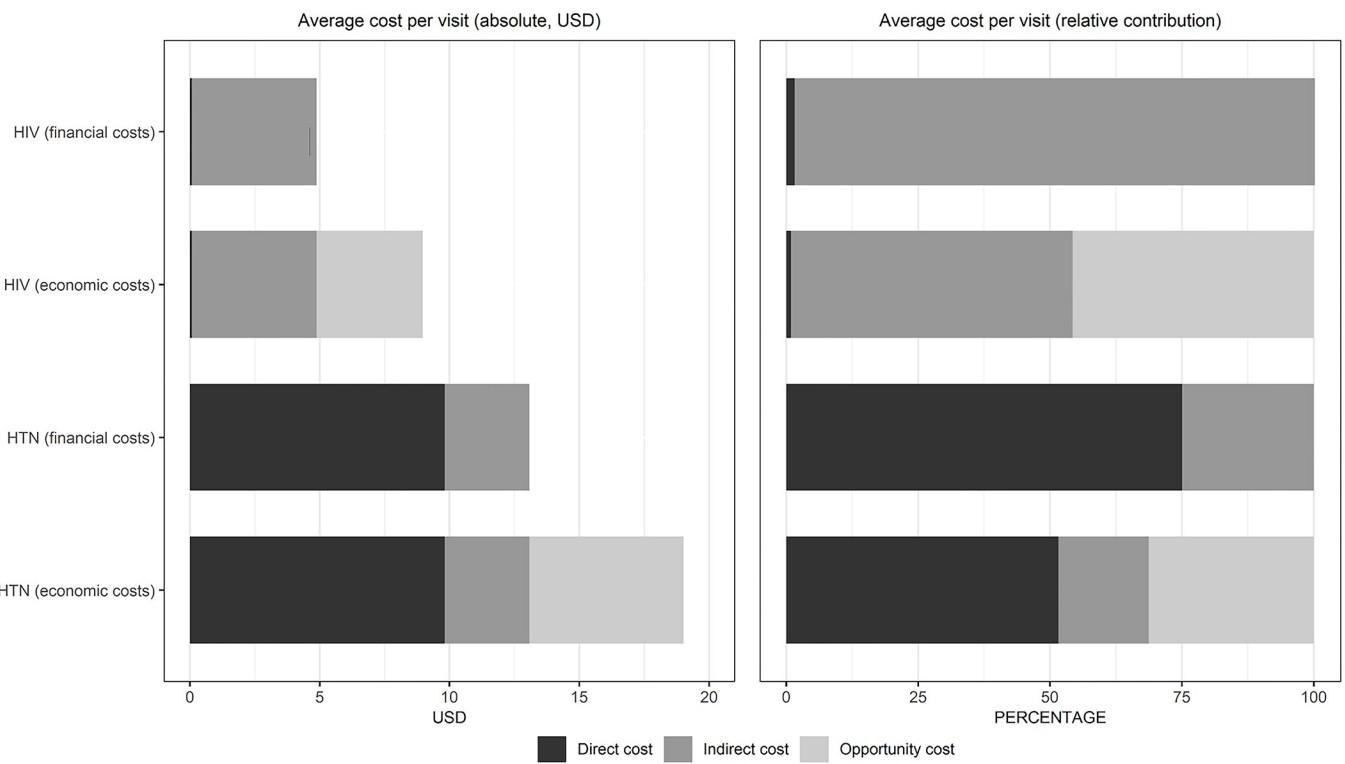

**Fig 1. Average cost per visit for HIV and HTN care (absolute vs. relative contributions).**

costs comprised the largest category of indirect costs for all patients. Median total indirect HIV care costs were lower under full lockdown ($2.48) compared to under partial lockdown ($3.80), largely due to a difference in transport costs. Observed mean differences in indirect costs were also statistically significant (p < 0.01) (see: **S1 Table**). When including opportunity costs into the total cost of patient care, median economic costs totaled $6.43 USD for participants. Median opportunity costs were lower for patients under full lockdown ($4.24 USD) than they were for patients under partial lockdown ($9.21 USD), however the mean difference in these opportunity costs was not statistically significant (see: **S1 Table**).

For hypertension, the median total financial cost per visit for patients was $5.70 USD. Patients sampled during the full lockdown spent significantly less in terms of direct costs ($0.00 USD) compared to those sampled after partial lifting ($4.38 USD). The mean difference in direct costs was also statistically significant (p<0.10) (see: **S1 Table**). Patients under full lockdown reported spending a median of $2.27 USD less for indirect costs related to their HTN visit than did patients sampled later. The difference was also statistically significant (p < 0.05) when examining means (see: **S1 Table**), largely driven by transportation costs. Total economic costs of HTN care for patients averaged $6.65 USD for all patients.

Though not reported in **Table 3**, we also examine the total per-visit hypertension care costs according to whether patients reported visiting the same HIV facility for their last HTN care visit (n = 24) versus a different HTN facility (n = 70) for both financial (direct + indirect) and economic (financial + opportunity) costs. We find that for financial costs, the median per-visit cost for HTN care among those who report using the same HIV facility is $7.09 (IQR: $4.24-$10.43) while those using different facilities report a per-visit cost of $5.26 (IQR: $1.46-$18.71). For economic costs, the median per-visit cost for HTN care among those who report using the same HIV facility is $8.40 (IQR: $4.97-$11.98) while those using different facilities

report a per-visit cost of $6.28 (IQR: $1.46-$23.38). Using student's t-test, we find that mean differences are flipped (same-facility visits are cheaper on average than different facility visits) but that neither mean difference is statistically significant. Further, we lack sufficient data to examine whether patients who reported using the same HIV facilities for their HTN care combined any of their care visits with HIV appointments.

Using patient-reported data on projected number of monthly visits, **Table 4** estimates the monthly costs of care for either condition, as well as examines spending as a share of either household income or expenditure. Here, we report on median costs to account for outlier reports of very high care costs, or 3 or more visits per month, though an analogous table of mean values can be found in **S2 Table**. The median patient indicated that they planned to attend the HIV clinic 0.7 times every month (or 8.4 times annually) and 1.0 times per month for HTN visits (or 12 visits annually). Among patients who reported household income (n = 83), considering financial (col 2) or economic costs (col 3) respectively, HIV care visit

**Table 4. Median monthly care costs, projected visits, and hospitalization costs.**

| | Median [IQR] | n | *Financial Costs* | | *Economic Costs* | |
|---|---|---|---|---|---|---|
| | | | Median [IQR] | n | Median [IQR] | n |
| | | | | | | |
| *Monthly HIV care* | | | | | | |
| Projected number of monthly visits | 0.7 | 94 | | | | |
| | [0.7–0.7] | | | | | |
| Median monthly total HIV care costs (USD) | | | $2.24 | 94 | $3.90 | 94 |
| | | | [$1.02 - $4.68] | | [$1.56 - $8.09] | |
| Median monthly HIV costs as % of HH income | | | 2.7% | 83 | 4.0% | 83 |
| | | | [1.0% - 5.0%] | | [2.2% - 8.2%] | |
| Median monthly HIV costs as % of HH expenditure | | | 1.8% | 82 | 3.9% | 82 |
| | | | [1.0% - 5.6%] | | [1.4% - 88.0%] | |
| *Monthly HTN care* | | | | | | |
| Projected number of monthly visits | 1.0 | 94 | | | | |
| | [0.7–1.3] | | | | | |
| Drug costs (USD) | $3.70 | 92 | | | | |
| | [$1.50 - $8.98] | | | | | |
| Median monthly total HTN care costs (USD) | | | $7.36 | 94 | $8.96 | 94 |
| | | | [$3.02 - $17.54] | | [$3.56 - $20.65] | |
| HTN costs as % of HH income | | | 7.1% | 83 | 7.9% | 83 |
| | | | [3.5% - 18.1%] | | [4.1% - 22.3%] | |
| HTN costs as % of HH expenditure | | | 7.2% | 82 | 8.6% | 82 |
| | | | [3.0% - 18.0%] | | [3.4% - 21.7%] | |
| *Hospitalization (HTN)* | | | | | | |
| % hospitalized for HTN in last 12 months | 9.6% | 94 | | | | |
| Number of hospital stays in last 12 months | 1.0 | 9 | | | | |
| | [1.0–1.0] | | | | | |
| Per hospitalization stay cost (USD) | $43.84 | 9 | | | | |
| | [$39.46 - $102.30] | | | | | |

Notes: "Economic costs" category (col 3) includes foregone wage earnings due to HIV or HTN visit. "Financial cost" totals do not include foregone wage earnings from medical visits; Interquartile ranges (IQR: 25th– 75th percentiles) are reported in brackets;; Monthly HTN care costs include hospitalization and drug costs, monthly HIV care costs do not.

costs comprised between 2.7% and 4.0% of median monthly HH income. Monthly HTN costs were larger, comprising between 7.1% (IQR: 3.5%– 18.0%) and 7.8% (IQR: 4.1%– 22.3%) of patients' median monthly HH income. Notably, the 75th percentile of households devoted as much as a fifth of HH income to HTN care, with overall mean shares of HH income ranging between 16.9 to 22.4% (see: **S2 Table**). Monthly drug costs comprised half of monthly HTN care financial costs (median of $3.70 USD). Hospitalization costs attributable to HTN were reported for 9.6% of the patient sample and reached a median of $43.84 USD per hospitalization stay. Adjusted over the course of a year, annual hospitalization costs alone constitute 9.8 percent of those patients' median monthly income. Though not reported in the table, 28.7% of respondents (n = 27) report borrowing money or selling assets to cover the cost of HTN care in the last 12 months, at a median value of $58.46 USD (IQR: $29.23 –$210.44).

**Table 5** shows the HIV out-of-pocket cost results of the GLM specifications estimating predictors of monthly cost among patients. Across all model specifications for both HIV outcomes, we found that a) participants sampled under full COVID-19 lockdown spent significantly less than those sampled after and that the magnitude of this spending was greater in the financial compared to the economic models, b) overall spending was significantly higher with increased (log) number of monthly visits, and c) for those using a private motorcycle as their primary means of transport, relative to walking or biking. In our financial cost models (Model 1) we additionally found that travel time and every mode of transport (relative to walking and biking) were significant predictors of total cost, whereas in our economic models (Model 2), we found significant predictive power only from private motorcycle transport. In our financial models we also found that having to arrange for child or elder care (so the patient can attend visits) and travel time to the HIV facility were consistently positive predictors of cost. We did not find any relationship between demographic covariates and spending in model 1. In model 2, we separately found that giving up wages was a strong positive predictor of monthly economic costs of care and that travel time to the clinic was positively associated with monthly costs but became insignificant after adding demographic covariates to the model. We also found that age and gender (male) were significant positive predictors of spending. Overall, likelihood ratio tests (comparing adjusted and unadjusted models) along with Akaike (AIC) and Bayesian (BIC) information criteria suggest that our adjusted specifications produce good predictive power.

**Table 6** shows the GLM results for predictors of monthly out-of-pocket financial and economic costs of hypertension care for our sample. Across the two models (economic and financial costs), we find several significant predictors of out-of-pocket costs of HTN care, particularly in our adjusted models. In both models, being sampled under lockdown has a consistently negative relationship with OOP spending on HTN care and is significant in three out of four model specifications. We also find that number of monthly care visits, travel time to facilities, and whether patients visit different HTN facilities from the HIV clinics at which they were sampled all show a positive relationship with out-of-pocket spending (again, significant in three out of four model specifications). We also find that arranging child/elder care to attend HTN visits has a negative relationship with out-of-pocket costs in both economic and financial models, but that this relationship is only significant in adjusted financial and unadjusted economic models. Being hospitalized overnight for HTN care has a positive and significant relationship in three of four models presented, but this relationship becomes negative and significant after adjusting for demographics and foregone income in our economic specifications.

In terms of demographic predictors from our adjusted models, we find that increased age, and marriage are consistently significant positive predictors of out-of-pocket spending in both models. Meanwhile, greater household expenditure is associated with less out of pocket

**Table 5. GLM models for predictors of monthly HIV costs.**

| VARIABLES | Direct & indirect HIV costs (financial) | | | | Direct, indirect HIV costs & lost income (economic) | | | |
|---|---|---|---|---|---|---|---|---|
| | Model 1 | | Model 1 (adj.) | | Model 2 | | Model 2 (adj.) | |
| | Coef. | (se) | Coef. | (se) | Coef. | (se) | Coef. | (se) |
| Total monthly HIV care costs$^\$$ | | | | | | | | |
| Survey conducted under lockdown | -1.14*** | (0.26) | -1.02*** | (0.31) | -0.91** | (0.41) | -0.60** | (0.26) |
| Number of monthly HIV facility visits (*ln*) | 1.45*** | (0.22) | 1.33*** | (0.22) | 1.04*** | (0.24) | 0.91*** | (0.18) |
| Patient admitted overnight for HIV care (last 12 mo.) | -0.01 | (0.24) | 0.20 | (0.45) | -0.37 | (0.38) | -0.11 | (0.41) |
| Arranged (child/adult) care to attend HIV visit | 0.74*** | (0.17) | 0.34* | (0.20) | -0.03 | (0.23) | -0.18 | (0.17) |
| Travel time to/from HIV facility (hours) | 0.28*** | (0.04) | 0.21*** | (0.04) | 0.14*** | (0.05) | 0.07 | (0.06) |
| Mode of transport to HIV facility | | | | | | | | |
| *Walking / biking (reference)* | - | - | - | - | - | - | - | - |
| *Matatu* | 1.03*** | (0.37) | 1.02* | (0.59) | 0.12 | (0.51) | 0.19 | (0.32) |
| *Car* | 3.48*** | (0.92) | 3.01*** | (0.98) | 1.50 | (0.95) | 0.73 | (0.52) |
| *Private motorcycle* | 1.78*** | (0.42) | 1.48*** | (0.56) | 1.68*** | (0.62) | 1.15*** | (0.26) |
| *Boda-boda* | 2.30*** | (0.45) | 1.91*** | (0.52) | 0.62 | (0.50) | 0.28 | (0.41) |
| *Multi-modal* | 1.50*** | (0.44) | 1.27*** | (0.48) | 0.07 | (0.45) | -0.15 | (0.27) |
| Patient gave up wages to attend HIV visit | | | | | 2.04*** | (0.78) | 1.30*** | (0.48) |
| *Demographics* | | | | | | | | |
| Age (years) | | | 0.01 | (0.01) | | | 0.02* | (0.01) |
| Male | | | 0.18 | (0.39) | | | 0.50** | (0.20) |
| Patient has secondary education or better | | | -0.05 | (0.13) | | | -0.08 | (0.18) |
| Patient is employed | | | 0.15 | (0.14) | | | -0.05 | (0.23) |
| Monthly HH expenditure (per 100 USD) | | | -0.01 | (0.09) | | | 0.01 | (0.03) |
| Patient is married | | | 0.33 | (0.25) | | | 0.04 | (0.19) |
| Facility fixed effects | Yes | | Yes | | Yes | | Yes | |
| Constant | -0.86 | (1.07) | -0.76 | (2.02) | -0.48 | (0.91) | -0.04 | (1.01) |
| AIC | 4.39 | | 4.25 | | 5.61 | | 5.22 | |
| BIC | -4.54E+01 | | -8.63E+01 | | 6.26E+02 | | 2.67E+02 | |
| Likelihood Ratio Test$^{\$\$}$ | | | 0.0003 | | | | 0.0000 | |
| Observations | 94 | | 94 | | 94 | | 94 | |

Notes: *** p<0.01, ** p<0.05, * p<0.1; $^\$$Model 1 and adjusted model 1 include only direct and indirect costs of care reported by respondents within the dependent variable, Model 2 and adjusted model 2 additionally include lost income per visit reported by respondents within the dependent variable; $^{\$\$}$ Likelihood ratio tests report Prob > chi2, likelihood that model with additional covariates is nested in the prior model, improving fit, however LR tests assume that robust standard errors are not used (in tables results are reported using robust SEs); Robust standard errors in parentheses; 12 missing observations for household expenditure were replaced with median household expenditure for the sample; HIV costs in all models do not include additional costs incurred for any overnight/hospitalization care related to HIV.

spending on HTN care from both perspectives. We find conflicting directions of association with educational achievement (positive from an economic perspective, and negative from a financial perspective). We also find that being male is associated with greater spending and that being employed is negatively associated with spending, but only in our financial model. Again, likelihood ratio tests (comparing adjusted and unadjusted models) along with Akaike (AIC) and Bayesian (BIC) information criteria suggest that our adjusted specifications produce good predictive power. As a robustness check, we additionally tested the use of a generalized linear mixed-effects (GLMM) specifications for each of the 4 adjusted and unadjusted models, treating each of the clinics as a random effects modifier. Our results did not differ meaningfully from those reported in Tables 5 and 6.

**Table 6. GLM models for predictors of monthly HTN care costs.**

| VARIABLES | Direct & indirect HTN costs (financial) | | | | Direct, indirect HTN costs & lost income (economic) | | | |
| --- | --- | --- | --- | --- | --- | --- | --- | --- |
| | Model 3 | | Model 3 (adj.) | | Model 4 | | Model 4 (adj.) | |
| | Coef. | (se) | Coef. | (se) | Coef. | (se) | Coef. | (se) |
| Total monthly hypertension care costs$ | | | | | | | | |
| Survey conducted under lockdown | -0.46 | (0.71) | -2.37*** | (0.15) | -9.22*** | (0.22) | -2.95*** | (0.66) |
| Number of monthly HTN clinic visits (*ln*) | 0.60 | (0.55) | 8.38*** | (0.70) | 4.09*** | (0.13) | 3.14*** | (0.54) |
| Patient admitted overnight for HTN care (last 12 mo.) | 0.58* | (0.34) | 7.51*** | (0.62) | 2.02*** | (0.16) | -3.17*** | (0.97) |
| Arranged (child/adult) care to attend HTN visit | -0.46 | (2.20) | -4.96*** | (0.29) | -3.02*** | (0.27) | -0.35 | (0.45) |
| Travel time to/from HTN facility (hours) | 0.12 | (0.13) | 2.24*** | (0.18) | 0.52*** | (0.05) | 1.17*** | (0.20) |
| Patient visits different facility for HTN care | 1.16 | (0.90) | 13.67*** | (1.21) | 3.61*** | (0.13) | 3.30*** | (0.72) |
| Patient gave up wages to attend HTN visit | | | | | 7.09*** | (0.20) | -0.49 | (0.55) |
| *Demographics* | | | | | | | | |
| Age (years) | | | 0.38*** | (0.03) | | | 0.10*** | (0.03) |
| Male | | | 7.49*** | (0.80) | | | 0.09 | (0.48) |
| Patient has secondary education or better | | | -0.41*** | (0.03) | | | 1.88*** | (0.56) |
| Patient is employed | | | -4.92*** | (0.40) | | | -0.23 | (0.21) |
| Monthly HH expenditure (per 100 USD) | | | -2.34*** | (0.21) | | | -0.23*** | (0.03) |
| Patient is married | | | 0.76*** | (0.05) | | | 3.09*** | (0.54) |
| Facility fixed effects | Yes | | Yes | | Yes | | Yes | |
| Constant | 2.93*** | (0.59) | -26.04*** | (2.58) | -6.64*** | (0.37) | -3.52 | (2.29) |
| AIC | 8.52 | | 7.90 | | 8.98 | | 8.46 | |
| BIC | 1.92E+04 | | 9.12E+03 | | 3.07E+04 | | 1.63E+04 | |
| Likelihood ratio test$$ | | | 0.0000 | | | | 0.0000 | |
| Observations | 94 | | 94 | | 94 | | 94 | |

**Notes**: *** p<0.01, ** p<0.05, * p<0.1; $Model 3 and adjusted model 3 include only direct and indirect costs of care reported by respondents in the dependent variable, Model 4 and adjusted model 4 additionally include lost income per visit reported by respondents in the dependent variable; $$ Likelihood ratio tests report Prob > chi2, likelihood that model with additional covariates is nested in the prior model, improving fit, however LR tests assume that robust standard errors are not used (in tables results are reported using robust SEs); Robust standard errors in parentheses; 12 missing observations for household expenditure were replaced with median household expenditure for the sample in adjusted models; Unlike HIV models, HTN care costs in all models include hospitalization expenses among those who had an overnight visit or were hospitalized for a hypertension-related condition.

## Discussion

Our findings have several important implications for the literature on the burden of HIV and HTN costs among co-morbid patients in Uganda and similar settings, the potential for integration of HIV and HTN services, the growing evidence on the main drivers of these costs, and the sensitivity of these costs to acute socio-economic shocks like the 2021 full COVID-19 lockdown, discussed in-turn below. Non-extrapolated patient costs for HIV represent between 2.6 to 4.0% of median patient household income, depending on whether financial or economic costs are considered, respectively. These relatively low costs of HIV care are reflective of the high availability of cost-covered HIV care within Uganda's health system and free antiretroviral medication available through PEPFAR's legacy of support [22–24]. By contrast, the high recurring median costs of hypertension care we observe (between 7.1 to 7.8% of median household income) are likely a significant determinant of low uptake of hypertension treatment in Uganda, where, by one estimate, less than a quarter of individuals diagnosed with hypertension are currently taking medication for treatment [25]. Additionally, for a subset of patients sampled with hospitalization expenses (9.6% of our sample), we find that overall HTN costs are catastrophic.

Spending as a percent of household income within our sample is generally in line with the sparse but growing literature on patient costs for these diseases and conditions in east African countries [26]. In a study of patient costs at integrated health care facilities in Tanzania and Uganda, Shiri and colleagues reported that, respectively for standalone HIV and HTN services, the mean costs shouldered by patients (direct, indirect, and opportunity costs, not including hospitalization) represented 5.6 and 6.3 percent of household income [27]. Mnzava and colleagues (2018) found that mean HIV care costs for patients represented about 7.9 percent of household income of the average Tanzanian [28]. Oyando and co-authors reported that mean patient costs for hypertension care (including hospitalization) exceeded 10 percent of household income for 59 percent of the sample of participants they interviewed at public health facilities in Kenya [29]. In our sample, 45.8% of participants report HTN costs exceeding this 10 percent of household income threshold (though we do not have data on public versus private facility spending). Finally, the share of patient costs attributable to direct healthcare spending, indirect spending to access care, and opportunity costs was 1%, 53%, and 45% for HIV while for HTN it was 52%, 17% and 31%, respectively.

This divergent share of direct costs for each disease may be explained in part by availability of services and medications. In Uganda HTN and HIV services are provided at no direct cost to patients at all public health facilities [30, 31]. However, while ART is readily available, anti-hypertensive medications are often out of stock in public facilities or patients' preferred medications are not available, with patients' only recourse to purchase medications that are not fully cost-covered from private pharmacies to maintain adherence [30, 31]. The high share of indirect expenditures and/or opportunity costs is in line with various contexts where these have been demonstrated to comprise significant shares of overall health-related expenditures for PLHIV [27–30, 32–37].

From the demand-side, these high overall costs, and the share of input cost drivers like indirect costs, and the co-constituent burden of the overall monthly number of visits for HIV and HTN care (a combined median of 20.4 annual care visits– 8.4 for HIV and 12 for HTN), imply potential cost savings from combining care visits. However, even by piggy-backing care visits for HTN within existing HIV visits, it is unclear how other out-of-pocket costs might change for patients shifting to integrated care. This further emphasizes the importance of trials like PULESA-Uganda that precisely measure overall costs and cost savings under these scenarios.

Though our out-of-pocket survey did not ask respondents to report the type of services received during clinic visits for hypertension, many visits were anecdotally reported to be made solely to pick up antihypertensive medications, which is consistent with prior findings in [38, 39]. Indeed, we find that the 90[th] percentile of number of monthly HTN visits is 3.3, with some patients in our sample reporting as many as 6–10 planned visits per month. Supportive to this narrative, recent evidence from Uganda suggest that the price and availability of hypertension medications can vary widely across public facilities over time and that patients often fill as little as half of the prescribed drugs from a given visit [38, 39]. This suggests further scope to reduce overall visits and/or implement differentiated service delivery (DSDM) schemes like community-based drug distribution and multi-month drug distribution, further limiting indirect and opportunity costs of care. Best case, attaching anti-hypertensive delivery to differentiated service delivery models for HIV in Uganda could offer additional reductions in patient costs. However, further study is needed.

Additional policy options, such as vouchers that help to offset paid forms of transportation (Matatus, boda-bodas and multi-modal transport) could also meaningfully reduce costs. Indeed, SEARCH, a recent integrated HIV-NCD integrated program in Uganda, provided transportation vouchers to patients (alongside a host of other patient-centered interventions), ultimately achieving significant gains in rates of patient participation along the continuum of

care [40, 41]. Though we have incomplete information on the current modes of transport for HTN services, our data on the significance of different modes of transport for HIV services (and the presumption that under integration services would be housed at the same facilities), as well as prior research on the outsized burden of transportation as a component of out-of-pocket costs of care suggests that transportation will remain a critical indirect cost input under integration [42–44].

Other cost predictors seem critical to any evaluation of the costs of HIV and HTN care. In our regression models, transport time is a reasonably consistent predictor of increased overall costs of monthly spending for both diseases. However, the impact of travel time alone has implications beyond pecuniary and lost income costs. In a systematic review of studies examining associations between travel time/distance to healthcare facilities and health outcomes, Kelly and colleagues (2016) found that those living farther away from care centers tend to experience worse health outcomes along several dimensions (e.g., survival rates, hospital stays, non-attendance at follow-up visits) [45]. Though our findings from a small sample suggest that efforts to bring care closer to patients (e.g., through community-based care delivery platforms) could improve out of pocket cost savings, benefits are likely to extend beyond pecuniary and even economic costs.

Additionally, in our adjusted HTN models, marital status is positively associated with spending. This may indicate that inter-household social dynamics are a factor in sustaining monthly spending and therefore higher quality care; though, one meta-analysis found that structural support systems such as marriage were not significantly related to improvements in adherence to hypertension treatment [46]. Thus, married patients may spend more on HTN services for other reasons not yet explored in the extant literature. Regardless, hypertension patients may stand to benefit from leveraging social mechanisms to improve salience and accountability, and opportunities exist to leverage peer-to-peer networks developed in HIV programs to improve care navigation and adherence in for PLHIV and HTN [46].

Importantly, we do find evidence that a small proportion of co-morbid patients are already engaging in some form of HTN care at some of the study's HIV facilities during our pilot. We do not find a significant difference in per-visit costs of HTN care among those who reported visiting the same HIV facilities for their HTN care compared to those reporting the use of different facilities. However, data to support this conclusion are incomplete–the small total number of patients (n = 24) and wide range of costs incurred increase the uncertainty of our estimates–and little is known about the overall quality or extent of care received at HIV facilities prior to a formal integration of services (i.e., whether "care" includes only BP screening/monitoring or actual treatment). Limited published evidence suggests that some screening/monitoring is already taking place at many HIV facilities across sub-Saharan Africa, but both screening *and* treatment are infrequently observed [47]. Compellingly, in our final regression models, we do find a consistently positive (and mostly significant) relationship between attending different facilities for HTN care than those where HIV care is sought and the monthly cost of HTN care. However, these findings do not include HIV costs nor reflect any adjustments for bundling care for different visit types. More evidence on clinically-ideal integration of services and the current standard of care is urgently needed.

The role of the 2021 full COVID-19 lockdown on HIV and HTN spending has several implications. First, we cannot identify whether observed cost differences during versus after the 2021 full COVID-19 lockdown are causal (i.e., due to the full lockdown itself), or if the full lockdown and its subsequent partial lifting created selection into care seeking behavior. Indeed, we do not observe any statistically significant demographic differences in the sample observed under full lockdown vs. after partial lifting (e.g., income, employment, education, expenditure, sex, marital status), which complicates this story. However, there are several

economically meaningful (if not statistically significant) differences in mean education, distance and travel time to facilities, and monthly household income and expenditure, which all point towards a selection narrative. Additionally, our sample is both small and not proportionally balanced by lockdown status, increasing standard errors in our estimated mean comparisons. Due to data limitations, we cannot test for other potentially meaningful differences across other characteristics (e.g., job type, household assets (i.e., wealth), or household size), and we do not observe health outcomes (e.g., whether HTN or HIV are controlled or uncontrolled).

Second, though significance of any relationship between care costs and lockdown status varied depending on disease condition and cost perspective, this relationship is consistently negative in all our regression estimates (and significant in seven of eight model specifications presented). For HIV care costs, the unadjusted per-visit cost of care was substantially lower under full lockdown from both a financial and an economic perspective, including after covariate adjustment. A significant difference in input costs seems to come from indirect costs of care. From a financial cost perspective, since there is no difference in direct costs of HIV care (HIV care is effectively free to all patients) and it is unlikely that indirect care costs (e.g., transport) would meaningfully decrease during a full lockdown period, we could intuit that those patients who either already faced (or could arrange) lower indirect costs of care were more likely to attend clinics. When we take an economic cost perspective, we see that opportunity costs were also lower under full lockdown. This might suggest that those who had fewer income generating opportunities may have been more likely to attend HIV clinics during full lockdown.

Third, for per-visit hypertension care, the magnitude of these differences by lockdown status was larger, but less precise. This imprecision for HTN costs is not unexpected since patients do not report whether the last visit for HTN care happened during or outside of the full lockdown period. A significant contributor to this per-visit cost was the reduction in costs associated with transportation where median transport costs were $0.00 during the full lockdown and $0.73 during the partial lockdown (a mean difference of $1.02 USD reported in **S1 Table**, $p < 0.05$). Evidence suggests that transportation fares were reported to be higher during prior lockdowns in Uganda, so this finding additionally suggests possible selection into care among those for whom transportation costs were normally less expensive (e.g., those who lived closer to facilities), or that patients found alternative modes of transportation (e.g., walking, biking, riding with a friend) [48]. Indeed, we do find both distance in kilometers (for HIV) and travel time (for both HIV and HTN visits) were slightly less for those sampled during full lockdown, though these differences were not statistically significant. We have no such data for HTN visits, but we also find significant differences in HIV visit transport mode during full lockdown. According to our data, respondents were more than three times less likely to take *matatus* (or mini-busses, typically a longer-distance form of transport that was severely curtailed during full lockdown) and more than 7 times less likely to use multi-modal paid transport to attend HIV visits (including *matatus*, private cars and motorcycles, and *boda-bodas*, which may be more likely for patients traveling further distances), while being 1.6 times more likely to only take *boda-bodas* and nearly 5 times more likely to walk or bike [19].

Taken together, the evidence suggests that monthly spending on disease management including regular care visits appears to be highly sensitive to economic shocks like that experienced under the 2021 COVID-19 full lockdown. Qualitative evidence from the 2020 lockdown in Uganda shows that patients experienced medicine stockouts (especially in rural areas) and that public transportation was less available—which combined could lead to lower spending [48, 49]. Unfortunately, our data does not allow us to estimate how this shock may have affected management of either disease, though evidence from prior COVID-19 lockdowns has

shown negative impacts on health and access to healthcare services [48, 50]. With these caveats, it is easy to argue that patients would likely benefit from disease management plans for either condition that allow them to weather such shocks, such as stockpiling needed medications through multi-month dispensing, providing differentiated service delivery, or otherwise limiting the need for greater frequency of care visits.

## Strengths and limitations

To our knowledge, this study is among the first to empirically examine predictors of patient out-of-pocket costs of HIV *and* HTN care in Uganda, doing so from multiple perspectives. We also provide novel evidence on the relationship between a COVID-19 lockdown and out-of-pocket spending for HIV and NCD care, and the first such evidence on the 2021 Uganda lockdown, in-particular. Our findings are also timely in helping to inform a rapidly changing policy environment for NCD care in sub-Saharan Africa. However, several factors must temper our results.

First, compared to other studies of out-of-pocket costs for HIV and HTN in sub–Saharan Africa, our sample population is relatively small and relies exclusively on patient self-report, which adds both imprecision and potential bias to our cost estimates. Second, though our patient population may be proportional to the number of co-morbid patients enrolled in facilities from which they were sampled and represent a decent cross section of medium-to-large HIV facilities (i.e., those serving >400 patients per year) in the country, we caution that the subset of patients who took the OOP survey were drawn from a larger convenience sample of patients selected for the KAP survey. Thus, we cannot confidently generalize these results to a larger population or determine if our population is systematically different from those for whom HTN status is unknown. Third, we do not observe patient medical data on management or status of either disease condition (i.e., whether the patient has controlled or uncontrolled HIV or HTN or other co-morbidities). We also did not observe any data on changes in how patients sought care during full lockdown that may be cheaper than traditional modes (e.g., shifting to online or virtual consultations). Such a shift may have led to a more optimal decrease in care costs that might not correspond with a resulting decrease in care quality, which we would not capture. Therefore, the out-of-pocket patient costs reported here may over- or (more likely) under-represent the true cost of *proper* disease management of either condition under full compliance of recommended treatment and/or may omit other important medical factors.

Fourth, we have limited data on the HTN services consumed by patients in our sample such as location and type of facility (public vs. private), how recently different kinds of HTN care (diagnosis, consultation, blood testing, medication purchases, etc.) were received, and whether the full range of HTN care services were obtained at one or across multiple locations. Fifth, due to a data collection error, we also lack disaggregated direct input cost data for HTN services for the whole sample, making it impossible to determine which inputs (e.g., personnel, medications, service fees, etc.) may be driving observed differences in direct cost. Sixth, although we took steps to achieve balance in terms of the size of facilities studied, only two of the 10 HIV facilities from which we sampled patients were private/not-for-profit. This lack of variation, combined with limited direct input cost data, lessens our ability to examine whether direct costs for either condition meaningfully differ according to facility type, and our evaluation of the predictors of care costs. Indeed, previous studies have found substantial differences in cost according to facility type for direct costs like medication and user fees, and more research on this topic is needed [38, 51, 52].

Seventh, some of the large input costs of care we observe from hospitalization for either HIV- or HTN-related conditions may be in part due to patients who live far from these facilities (in remote rural areas) and require overnight hospital care because being discharged early was not feasible, or imminent returns to the facility for follow-up care or complications would be necessary. Unfortunately, we cannot evaluate this hypothesis as our data on the facilities where patients were hospitalized for either condition is incomplete. For example, it may not be reasonable to assume that the same facilities at which patients receive regular (or their last) HTN or HIV care would be the same ones utilized for overnight hospitalization. Finally, this analysis is limited in that we utilize cross-sectional data to report on costs at a given moment in time for patients and, given our small sample sizes and limited set of covariates, we do not predict cost estimates through our regression analyses to inform policy. Instead, our analyses should be seen as exploratory and hypothesis generating.

## Conclusion

The patient costs of HIV and HTN care in Uganda are substantial. Large potential out-of-pocket savings could be realized by integrating care services for HIV and HTN in place, person, and time, though further study from large, well-identified trials like PULESA-Uganda are urgently needed to test this hypothesis.

## Supporting information

**S1 Checklist. Inclusivity in global research.**
(DOCX)

**S1 Appendix. GLM data procedures.**
(DOCX)

**S1 Table. Mean cost (USD) per visit breakdown.**
(DOCX)

**S2 Table. Mean monthly care costs, projected visits, and hospitalization costs.**
(DOCX)

## Acknowledgments

We wish to thank without implicating Isaac Ssinabulya, Moses R. Kamya, and Donna Spiegelman for their support to the completion of this manuscript.

## Author Contributions

**Conceptualization:** Faith Kagoya, John Baptist Kiggundu, Brian Hutchinson, Jeremy I. Schwartz, Gerald Mutungi, Rachel Nugent, David Contreras Loya.

**Data curation:** Faith Kagoya, John Baptist Kiggundu, Robert Twine, Martin Muddu, Gerald Mutungi, James Kayima, Anne R. Katahoire.

**Formal analysis:** Drew B. Cameron, Lillian C. Morrell, Faith Kagoya, Brian Hutchinson, Robert Twine, David Contreras Loya.

**Funding acquisition:** Chris T. Longenecker, Fred C. Semitala.

**Methodology:** Drew B. Cameron, Lillian C. Morrell, David Contreras Loya.

**Project administration:** Faith Kagoya, John Baptist Kiggundu, Robert Twine, Martin Muddu, Gerald Mutungi, James Kayima, Anne R. Katahoire.

**Supervision:** Drew B. Cameron, Brian Hutchinson, Robert Twine.

**Visualization:** Lillian C. Morrell.

**Writing – original draft:** Drew B. Cameron, Lillian C. Morrell, Faith Kagoya, John Baptist Kiggundu, Brian Hutchinson, Robert Twine, Jeremy I. Schwartz, David Contreras Loya.

**Writing – review & editing:** Drew B. Cameron, Lillian C. Morrell, Faith Kagoya, John Baptist Kiggundu, Brian Hutchinson, Robert Twine, Jeremy I. Schwartz, Martin Muddu, Gerald Mutungi, James Kayima, Anne R. Katahoire, Chris T. Longenecker, Rachel Nugent, David Contreras Loya, Fred C. Semitala.

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
