## [Decision Letter · Decision Letter 0]

29 Mar 2024

PGPH-D-24-00258

Out of Pocket Care Costs Among HIV and Hypertension Co-morbid Patients in Urban and Peri-Urban Uganda

Dear Dr. Cameron,

Thank you for submitting your manuscript to PLOS Global Public Health. After careful consideration, we feel that it has merit but does not fully meet PLOS Global Public Health’s publication criteria as it currently stands. Therefore, we invite you to submit a revised version of the manuscript that addresses the points raised during the review process.

This is an important study measuring the out-of-pocket care costs among patients living with HIV(PLHIV) and Hypertension in Urban and Peri-urban settings of Uganda, especially in the setting of universal health coverage.

Some of the points the authors should address to improve the quality of the paper are to give more clarity to the rationale of this study in connection to the PULESA-Uganda trial, to mention also the newer formats for reporting economic evaluation.

As has been suggested  rephrasing of some of the important points and also more careful with the language at  certain areas also will improve readability and acceptability of the manuscript

We look forward to receiving your revised manuscript.

Kind regards,

Suma Krishnasastry, MBBS, MD,DNB

Academic Editor

Journal Requirements:

Additional Editor Comments (if provided):

Reviewers' comments:

Reviewer's Responses to Questions

**Comments to the Author**

1. Does this manuscript meet PLOS Global Public Health’s publication criteria? Is the manuscript technically sound, and do the data support the conclusions? The manuscript must describe methodologically and ethically rigorous research with conclusions that are appropriately drawn based on the data presented.

Reviewer #1: Yes

Reviewer #2: Yes

2. Has the statistical analysis been performed appropriately and rigorously?

Reviewer #1: Yes

Reviewer #2: Yes

3. Have the authors made all data underlying the findings in their manuscript fully available (please refer to the Data Availability Statement at the start of the manuscript PDF file)?

Reviewer #1: Yes

Reviewer #2: Yes

4. Is the manuscript presented in an intelligible fashion and written in standard English?

Reviewer #1: Yes

Reviewer #2: Yes

5. Review Comments to the Author

Reviewer #1: Overall comments:

This study measures the out-of-pocket care costs among patients diagnosed with HIV and Hypertension in Urban and Peri-urban settings of Uganda. The study addresses an important topic and fill the knowledge gap in this area; however, there are several aspects that could be clarified or strengthened throughout the paper. Therefore, I have made several specific comments and hope this will help the authors improve this paper's quality.

Specific comments:

Title:

1. The authors identified the type of costs (i.e., out of pocket care expenses) in the title, however it is not clear if these cost estimations were done retrospectively or focused a current state of illnesses (i.e., snapshot). It would be good to highlight this in the title, so readers get it at the very outset of the paper.

Abstract:

1. The authors need to add a primary "study objective” statement in the background.

2. The authors stated that respondents were sampled during the nationwide lockdown, but it is not clear what is the sampling cohort/frame of this study (for example, how many potential patients HIV/HTN are seen in 10 clinics on an average in a year)?

Introduction:

1.In line # 66 on Pg3, the authors need to provide the full forms for “HIV”, and “UNAIDS”.

2. I wonder if the term “PLHIV” is widely known (cited or common) for readers in this journal? If not, it may be better to use just HIV patients.

3. It seems like the PULESA-Uganda trial will take place in 16 facilities, however this paper reports on the survey undertaken in 10 clinics. Could this be clarified in the intro. or the methods section, if possible?

4.The rationale of this study in connection to the PULESA-Uganda trial could be misleading and is not supported by the evidence. In line # 111 on Pg5, the authors reported that findings from this cross-sectional survey will help to understand the potential efficiencies that might result from the integration. I find this statement problematic because the trial has not started yet, and how could authors make a claim that trial will result in fewer visits or improve efficiencies? It could be that trial findings are not statistically in favor to support this claim (or hypothesis). What if the trial finds more frequent visits than initially hypothesized? Anyway, I think this part needs to be reframed in a way that only focuses on understanding the costs, which is an important aspect to inform the subsequent economic model design and input parameters for the trial.

5. The authors need to add more literature on the context of HIV and HTN burden and care seeking practices in the introduction. At this point, it seems like they only provided information about the trial and how this study will link to it. The readers do not know much about the context of care seeking practices, so I would like to see that added to the intro. section.

Methods:

1. The authors did not use the standard format for reporting the economic evaluation. It is advisable to use (and cite) the CHEERs checklist in this paper (please see: https://www.ispor.org/heor-resources/good-practices/article/consolidated-health-economic-evaluation-reporting-standards-(cheers)---explanation-and-elaboration)

2. In line # 121 on page 5, the authors mentioned that survey was adapted from a previous study. Upon checking the reference (citation 17), it seems like this study reported on the financial costs of HIV, so it is not clear what aspects of the tool were adapted and what areas were added to the adapted version?

3. in line # 125 on page 5, could you please clarify what was the time frame for the recall? How did you define the last visit for care? Was that in the last year, two years, etc. from the date of survey? Please clarify the time point.

4. In line # 131 on page 6, the authors reported that facilities > 2000 patients were classified as large facilities. What is the basis for this criterion? Is that something based on the standard of care or assumption (please clarify)?

5. Table 1, It seems like all small facilities (<2000 patients) represent the public sector facilities.

I wonder, if you could reclassify the small facility size in way, say for instance, <3000 patients, you could have one small facility representation in the PNFP sector. Just a thought!

6. In line 145 on page 5, there is mention of sample size based on HTN prevalence, but I do not see any calculations or formulas showing how this was determined? Authors must provide the sample size calculation to demonstrate that 94 participants represent adequate power or precision.

7. The costs are reported as of 2022, so it would be good to use the World Bank’s GDP index of 2023/24 considering the readers may want to know the costs as of the current year.

8. In the analysis, the authors reported the application of generalized linear model. I think, there may be a clustering by the type and size of clinic, as some patients may be paying more or less depending on where they are treated. In this case, my suspicion is that clinic type/size may be playing a random effect modifier on the out of pocket costs. Therefore, I wonder if the authors could consider applying the generalized linear Mixed-Effect model (GLMM) instead?

11. Considering that costs were changing (between full and partial lockdowns), could the authors perform a sensitivity analysis to demonstrate how robust their cost estimates can be when assuming a 10%, 15% or 25% etc. variation in cost over time scenario?

Results:

1. In Table 2, could authors relabel the variable in column three “No lockdown” to “partial lockdown” based on how they defined in the methods section?

2. In Tables 2 and 3, considering the sample size in each lockdown and partial lockdowns are small, could authors report the estimates in median (and interquartile ranges-IQR) instead? It seems that standard error of means is flagging the need for medians and IQRs.

3. In Table 6, did authors consider a interaction term for variables ‘travel time to/from HTN facility’ and ‘admitted overnight’? It could be that patients who live far (in rural remote areas) had to stay in the hospital for an extra night or so, just because discharging them early was not feasible or they may have to return back due to disease complications etc.

Discussion:

1. Although it is understandable that fewer visits during the full lockdown may have contributed to lower costs, I wonder if there was any change in how patients sought care that may be cheaper than the traditional ways of seeking care? For example, online/virtual consultation with the care provider may be low cost, and prevented time/travel and missed opportunities? Could authors reflect on this in the discussion depending on the context of care in Uganda?

2. The section on “limitations” could be reworded as “Strengths and Limitations” to better align with the content in this section.

3. I think the statement (in line # 494), where authors report “while potential savings from integrating care at HIV……..of either condition” should be omitted or rephrased because it is not supported by any literature and the future trial findings cannot be assumed at this stage.

Reviewer #2: The manuscript is methodologically sound, with the data backing the conclusions. All data supporting the findings in the manuscript has been thoroughly provided. Authors are advised to enhance readability by revising the language

6. PLOS authors have the option to publish the peer review history of their article (what does this mean?). If published, this will include your full peer review and any attached files.

**Do you want your identity to be public for this peer review?** For information about this choice, including consent withdrawal, please see our Privacy Policy.

Reviewer #1: **Yes: **Asif Raza Khowaja

Reviewer #2: No

---

## [Editor Report · Decision Letter 1]

29 Jul 2024

Current out of pocket care costs among HIV and hypertension co-morbid patients in urban and peri-urban Uganda

PGPH-D-24-00258R1

Dear Dr. Cameron,

We are pleased to inform you that your manuscript 'Current out of pocket care costs among HIV and hypertension co-morbid patients in urban and peri-urban Uganda' has been provisionally accepted for publication in PLOS Global Public Health.

Best regards,

Suma Krishnasastry, MBBS, MD,DNB, FRCP (Edin)

Academic Editor